# Use of PVDF Wire Sensors for Leakage Localization in a Fluid-Filled Pipe

**DOI:** 10.3390/s20030692

**Published:** 2020-01-27

**Authors:** Pingling Sun, Yan Gao, Boao Jin, Michael J. Brennan

**Affiliations:** 1Key Laboratory of Noise and Vibration Research, Institute of Acoustics, Chinese Academy of Sciences, Beijing 100190, China; sunpingling17@mails.ucas.ac.cn (P.S.); jinboao16@mails.ucas.ac.cn (B.J.); 2University of Chinese Academy of Sciences, Beijing 100049, China; 3Department of Mechanical Engineering, UNESP, Ilha Solteira SP15385-000, Brazil; mjbrennan0@btinternet.com

**Keywords:** polyvinylidene fluoride (PVDF) wire sensor, fluid-filled pipe, leakage localization, pressure sensitivity

## Abstract

The detection and location of pipeline leakage can be deduced from the time difference between the arrival leak signals measured by sensors placed at the pipe access points on either side of a suspected leak. Progress has been made in this area to offer a potential improvement over the conventional cross-correlation method for time delay estimation. This paper is concerned with identifying suitable sensors that can be easily deployed to monitor the pipe vibration due to the propagation of leak noise along the pipeline. In response to this, based on the low-frequency propagation characteristics of leak noise in our previous study, polyvinylidene fluoride (PVDF) wire sensors are proposed as a potential solution to detect the pipeline leak signals. Experimental investigations were carried out at a leak detection pipe rig built in the Chinese Academy of Sciences. Their performances for leak detection were shown in comparison with hydrophones. It is suggested that with special considerations given to aspects pertaining to non-intrusive deployment and low cost, the PVDF wire sensors are of particular interest and may lead to a promising replacement for commercial leak noise transducers.

## 1. Introduction

High quality municipal utilities are fundamental to sustaining modern life and economic growth. Recently, a great challenge involving water loss in the stock of water distribution network, has confronts China’s buried infrastructure [1]. According to World Bank estimates, the global water losses reach 32 billion cubic meters each year, half of which occur in developing countries [2]. Detection and repair of pipeline leaks are critical functions of system operation and maintenance due to the increasing water scarcity and degradation. The problems associated with pipeline leakage are not unique to China and have led to renewed international awareness [3]. Historically, water distribution in China is extremely unbalanced among regions and seasons. Some cities suffer severe water stress, with the leakage rates reaching as high as 70%, far exceeding the national accepted standard of 12% [1,4]. Undoubtedly, improvements of the performance of the water distribution pipes for effectively transporting a water resource will provide significant social, environmental, and economic benefits. With this in mind, there is an increasing demand for the development of leak detection equipment for the accurate location of pipeline leakage on a cost-effective basis.

Leak localization is the identification of the position of a suspected leak prior to excavation and repair, which is normally coordinated in parallel to leakage management, i.e., monitoring and control. Since the second half of the nineteenth century, many leak detection techniques have been developed, primarily including acoustic, chemical tracers, infrared thermography, ground-penetrating radars, and in-line leak detectors [5,6,7,8,9]. Discussions of the technologies are detailed in [10,11]. The current imitations of non-acoustic methods are that they are more expensive, complex and time-consuming, which potentially restrict their applications in practical leak detection surveys. As such, acoustic methods continue to be widely used for leak detection and location in water distribution networks [12,13,14,15]. 

It is now common practice that the acoustic methods based on cross-correlation are relatively effective for pinpointing leaks in metallic pipes; however, difficulties are often encountered when applied in plastic pipes, even in an acoustically quiet environment, since leak noise in plastic pipes attenuates more heavily at lower frequencies. Substantial research has been carried out by the present authors for leakage localization in buried plastic pipes, with attention paid to improvement over the conventional cross-correlation method for time delay estimation (TDE) [13]. Moreover, it has been found that the application of the cross-correlation methods can be largely affected by the selection of leak noise transducers [12]. Duan et al. [16] carried out a feasibility study on leak detection in viscoelastic pipe systems using the frequency response function method. This work suggests that the transient input signals with repaid changes in time are preferable for leak detection in viscoelastic pipelines. Factors that affect the effectiveness of correlation techniques have been reviewed in the case studies of transmission mains and plastic pipe leak detection [17]. Early detection of leaks has been attempted in water-filled small-diameter plastic pipes by means of acoustic emission measurements and autocorrelation analysis of vibro-acoustic signals generated by simulated leaks [18,19]. 

Continuous (real-time) pipe network monitoring systems have commenced more recently to take the leak detection and location methods into a whole new sphere, which requires the deployment of a wireless sensor network to work steadily in extreme environments of high humidity and high corrosion [20,21,22]. Currently, there is a growing need for alternative leak noise sensors to fulfill commercial demands of cost, deployment, and reliability. Convectional leak noise transducers, namely accelerometers and hydrophones, are designed and manufactured based on the piezoelectric effect. Polyvinylidene fluoride (PVDF) is seen as an emerging piezoelectric material for manufacturing the leak noise transducers. The strong piezoelectric effect of PVDF was first discovered and proposed by Kawai [23], and the piezoelectric film made of this material has been widely used in transducer manufacturing techniques due to its high sensitivity and flexibility [24]. For example, PVDF piezoelectric film/cable has been applied in the manufacture of hydrophones (underwater acoustic transducers) and has proved to be highly sensitive to acoustic pressures [25,26]. 

In this paper, PVDF wire sensors are developed and their performance for pipeline leakage detection and location are assessed. It is organized as follows. Section 2 presents the principle of the PVDF wire sensor for monitoring pipe vibration induced by water leaks, along with its piezoelectric properties. The experimental rig and measurement procedures are given in Section 3, followed by the discussions of leak noise measurements using both the PVDF wire sensors and hydrophones in Section 4. Finally, some conclusions are drawn in Section 5.

## 2. Methodology

### 2.1. Overview of Leak Noise Propagation in a Fluid-Filled Pipe

Acoustic leak detection equipment operates at low frequencies, where the axisymmetric (n = 0) fluid-borne (s = 1) wave is responsible for propagation of leak noise [27]. Gao et al. [28] investigated the loading effects of the surrounding medium on the low-frequency propagation characteristics of the s = 1 wave. To offer the theoretical basis for leak noise measurements, the relationship between the internal acoustic pressure and the pipe wall displacements is briefly discussed with more detailed derivation in [28,29]. Figure 1 shows a fluid-filled pipe surrounded by an elastic medium. Referring to the figure, u, v, and w are the axial, circumferential, and radial displacements of the pipe, respectively, with mean radius a and wall thickness h such that h/a≪1 (i.e., thin-walled).

As suggested in [27], the traveling wave solutions were adopted in the analysis of the couple axial and radial motion of the fluid-filled pipe system. For the s = 1 wave, the acoustic pressure was found to be related to the pipe radial displacement by [30].
(1)$$P_{1}=\frac{\omega^{2}\rho_{f}}{k_{1}^{r}{J^{\prime }}_{0}\left(k_{1}^{r}a\right)}W_{1},$$
where $W_{1}$ and $P_{1}$ are the amplitudes of the pipe radial displacement and the acoustic pressure, respectively, $k_{1}^{r}$ is the radial wavenumber of s = 1 wave, ${\left(k_{1}^{r}\right)}^{2}=k_{f}^{2}-k_{1}^{2}$; $k_{1}$ is the s=1 wavenumber; $k_{f}$ is free-field fluid wavenumber, $k_{f}^{2}=\omega^{2}\rho_{f}/B_{f}$; $\rho_{f}$ and $B_{f}$ are the density and bulk modulus of the fluid, respectively; $J_{0}\left(\;\right)$ represents a Bessel function of order zero; and $J_{0}^{\prime }\left(\;\right)=\partial /\partial rJ_{0}\left(\;\right)$. At low frequencies when $k_{1}^{r}a\to 0$, applying the approximation for the Bessel function $J_{0}\left(k_{s}^{r}a\right)/\left[k_{s}^{r}aJ_{n}^{\prime }\left(k_{s}^{r}a\right)\right]\approx -2/{\left(k_{s}^{r}a\right)}^{2}$, Equation (1) leads to
(2)$$P_{1}J(k_{1}^{r}a)=\frac{-2B_{f}}{1-{\left(k_{1}/k_{f}\right)}^{2}}\frac{W_{1}}{a}.$$

For an elastic surrounding medium, the coupled equation of motion dominated by the s = 1 wave can be obtained by [27].
(3)$$\left[\begin{array}{cc} \Omega^{2}-{\left(k_{1}a\right)}^{2}-SL_{11} & -i\upsilon (k_{1}a)-SL_{12}\\ -i\upsilon (k_{1}a)-SL_{21} & 1-\Omega^{2}-FL-SL_{22} \end{array}\right]\left[\begin{array}{c} U_{1}\\ W_{1} \end{array}\right]=\left[\begin{array}{c} 0\\ 0 \end{array}\right],$$
where Ω is the non-dimension frequency, $\Omega =k_{L}a$; $k_{L}$ is the compressional wavenumber of the shell, $k_{L}^{2}=\omega^{2}\rho_{p}\left(1-\nu^{2}\right)/E_{p}$; $\rho_{p}$, ν, $E_{p}$ are the density, Poisson’s ratio, and Young’s modulus of the pipe wall, respectively; and FL and SL are the load terms representing the coupling effects due to the internal fluid and the surrounding medium, respectively, as defined in [27]. Setting the determinant of the characteristic matrix in Equation (3) as 0, the s = 1 wavenumber can be obtained by
(4)$$k_{1}^{2}=k_{f}^{2}\left(1+\frac{\beta }{1-\Omega^{2}+\alpha }\right),$$
where *α* and β are the measures of the loading effects of the surrounding medium and the internal fluid on the pipe wall; $\beta =2B_{f}a\left(1-\nu^{2}\right)/\left(E_{p}h\right)$; $\alpha =-\nu^{2}$ and $\alpha =-\nu^{2}-SL_{22}$ are the air and fluid medium, respectively, $SL_{22}=-\rho_{m}a\Omega^{2}H_{0}\left(k_{d1}^{r}a\right)/\left[\rho_{p}hk_{d1}^{r}aH_{0}^{\prime }\left(k_{d1}^{r}a\right)\right]$; $\rho_{m}$ and $\rho_{p}$ are the densities of the surrounding media and the shell; $k_{d1}^{r}$ is the radial wavenumber of surrounding media; $H_{0}\left(\;\right)$ is the Hankel function of the second kind and zero order; and $H_{0}^{\prime }\left(\;\right)=\partial /\partial rH_{0}\left(\;\right)$. In general, the leak noise mainly propagated at lower frequencies where the effect of the surrounding medium could be neglected in the calculation of the propagation wave speed [12]. Note that it traveled slightly slower in the submerged water pipe compared to the in-air water pipe [28]. Furthermore, the complex wavenumber is obtained from Equation (4), accounting for loss within the pipe wall by a complex elastic modulus and given by
(5)$$k_{1}^{2}=k_{f}^{2}\left(1+\frac{2B_{f}a}{E_{p}h+i\eta E_{p}h}\right),$$
where η is the loss factor of pipe material. The real and imaginary parts of the wavenumber lead to the wave speed and wave attenuation, respectively. Here, the wave speed was of particular interest and is obtained by
(6)$$c=c_{f}{\left(1+\frac{2B_{f}a}{E_{p}h}\right)}^{-1/2}$$

By substituting Equation (4) into Equation (2), the relationship between the internal acoustic pressure and the pipe radial displacements is obtained by
(7)$$W_{1}=\frac{P_{1}a^{2}}{E_{p}h}\frac{1-\nu^{2}}{1-\Omega^{2}+\alpha }$$

### 2.2. Piezoelectric Property of the PVDF Wire Sensor

The piezoelectric property of the PVDF has been studied in previous research [23,31,32,33]. It is a piezoelectric plastic material that generates a charge or voltage when mechanically deformed. The PVDF wire sensor has the form of coaxial design, with the Piezo polymer being the dielectric between the center core and the outer braid, as shown in Figure 2. Due to its coaxial design, the PVDF wire sensor is self-shielded, allowing its use in a high Electromagnetic interference environment. In addition, the PVDF has good piezoelectric properties, for example, Kowbel et al. [34] found that its piezoelectric coefficient *g*_31_ is much higher than other piezoelectric materials. 

The performance of the PVDF wire sensor can be analyzed with a one-dimensional, lumped parameter approach, since the PVDF piezo film is very thin and the deformation in the direction of length is much larger than those in the directions of width and thickness. Thus, it is termed a wire sensor. The piezoelectric equation can be obtained by
(8)$$D_{3}=d_{31}\sigma_{1},$$
where $D_{3}$ is the electrical displacement; $d_{31}$ is the piezoelectric strain constant; and $\sigma_{1}$ is the stress applied in the relevant direction. Therefore, the charge *Q* generated by the PVDF wire sensor is given by [33]
(9)$$Q=d_{31}\sigma_{1}S=d_{31}E_{l}\varepsilon S,$$
where $E_{l}$ is the Young’s modulus of PVDF; ε is the strain; and *S* is the conductive electrode area. 

Since the PVDF wire sensor was wrapped around the pipe, the circumferential strain in the pipe wall is given by
(10)$$\varepsilon =W_{1}/a_{out},$$
where $a_{out}$ is the outer radius of the pipeline.

Substituting Equations (7) and (10) into Equation (9), the pressure sensitivity of the wire sensor (the charge per unit internal pressure) is given by
(11)$$Q/P_{1}=d_{31}a^{2}\frac{E_{l}S}{E_{p}ha_{out}}\frac{1-\nu^{2}}{1-\Omega^{2}+\alpha }$$

As can be seen from Equation (11), the pressure sensitivity was proportional to the winding number of the pipe. It was also governed by the pipe material. For typical in-air metal and plastic pipes, as suggested in [27], the properties are listed in Table 1. For the pipe with the same dimensions, the pressure sensitivity on the PVC pipe was calculated to be over 23 times the cast iron pipe. It was thus expected that good performance of PVDF wire sensors would be achieved in the leak noise measurements in the plastic pipes.

### 2.3. Pipe Leakage Localization

When a pipe leak occurs, the leak noise will travel in both directions along the pipeline through the pipe wall and the water, which can be captured by leak noise transducers installed on the pipe wall and inside the pipe bracketing the leak, as shown in Figure 3.

Referring to Figure 3, the leak location relative to one of the sensors, for instance $d_{1}$ can be calculated by
(12)$$d_{1}=\frac{D-cT_{0}}{2},$$
where *D* is the distance between the two sensors; *c* is the propagation speed of leak noise given by Equation (6); and $T_{0}$ is the estimated time delay between sensor signals. Given the distance *D* and the propagation speed *c,* the problem of leakage localization was now transformed into the estimation of time delay $T_{0}$. At low frequencies when the fluid wavelength was much larger than the pipe diameter, the amplitudes of acoustic pressures, $P_{1}\left(\omega \right)$ and $P_{2}\left(\omega \right)$, captured by the two sensors can be expressed as
(13)$$P_{1}(\omega )=P_{0}(\omega )e^{-ikd_{1}},P_{2}(\omega )=P_{0}(\omega )e^{-ikd_{2}}$$
where $P_{0}\left(\omega \right)$ is the amplitude of acoustic pressure at the leak location and *k* is the wavenumber given by Equation (4), which can be re-expressed as k=ω/c−iβω, with *c* and β being the propagation speed and attenuation factor of the leak noise, respectively. Thus, cross-spectral density (CSD) function of the two sensors signals is obtained by [14]
(14)$$S_{12}(\omega )=S_{ll}(\omega )e^{-\beta \omega D}e^{i\omega T_{0}},$$
where $T_{0}=\left(d_{1}-d_{2}\right)/c$; $S_{ll}\left(\omega \right)$ is the auto-spectral density (ASD) function of the leak source. Noting the phase spectrum, $\Phi_{12}\left(\omega \right)=\mathrm{Arg}\left\{S_{12}\left(\omega \right)\right\}=\omega T_{0}$, the time delay, $T_{0}$, can be calculated based on the slope of the phase spectrum, which is subsequently substituted into Equation (12) to determine the leak location.

## 3. Experimental Setup and Procedure

### 3.1 The Experimental Rig

The experiments were performed on a pipeline leak test rig constructed in the Institute of Acoustics, Chinese Academy of Sciences, as shown in Figure 4. It was consisted of a polyethylene (PE) water pipe with a length of 24 m (55 mm radius, 5 mm thickness). Valves were installed at the inlet and outlet of the pipe rig. A simulated leak hole in the middle of the pipe can be changed by copper covers with different aperture sizes. A test pipe section of 6 m shown shaded in the figure was placed in the water tank. Five round holes on this section were drilled for the access of the B&K 8103 hydrophones (equidistant). PVDF wire sensors of two-turn (2t), four-turn (4t), and six-turn (6t) were used in the measurements. Photographs illustrating the sensor installation and data acquisition system are shown in Figure 5 and Figure 6, respectively. The leak signals of 60 s were collected by the B&K PULSE3050 acquisition system with a sampling frequency of 8192 Hz. High-pass filtering operation was conducted on the measured leak signals prior to the analysis to eliminate the deteriorating effect of background noise below 50 Hz, as suggested in [35]. 

### 3.2 Leak Noise Measurements 

Two sets of measurements were conducted to evaluate the performance of PVDF wire sensors in comparison with the hydrophones on the in-air and in-water pipe rig. Pressure sensitivities of the PVDF wire sensors were first measured to ensure that they were effective for leakage localization. It must be pointed out that wave behavior of the leak noise is quite complicated due to effects of the supports, as illustrated in Figure 7a. The support effects are obvious for the in-air pipe, which in turn affect the propagation of acoustic waves and cause leak localization errors. This will be demonstrated further in the data analysis. To overcome this problem, the test section was submerged at a depth of 5 cm (counting above the pipe), as shown in Figure 7b. The water pressure in pipe was 0.2 MPa and the water flow was 0.43 $m^{3}/h$. The leak hole had a diameter of 3 mm. For the measurements of pressure sensitivity, referring to Figure 5a, three PVDF wire sensors with 2t, 4t, and 6t, were installed close to the hydrophone at a distance of 0.9 m from the leak hole. For leak localization measurements, a pair of hydrophones were installed at the distances of 0.1 m and 0.9 m from the leak hole (and bracketing it). A pair of PVDF wire sensors with 6t were each installed next to them with a small gap of 3 cm, as shown in Figure 5b. To avoid the relative movements between the PVDF wire sensors and the pipe (i.e., slippage), PVDF wire sensors were installed as follows: I) apply strong double-sided adhesive tape to the pipe wall; II) wrap PVDF wire sensor at the position of the double-sided adhesive tape to secure the strong connection between the wire sensor and the pipe wall; III) wrap aluminum tape around the sensors to ensure water resistance (particularly important for the submerged pipe). 

## 4. Data Analysis

### 4.1. Pressure Sensitivity of PVDF Wire Sensors 

Consider the water pipe in-air. To evaluate the performance of the wire sensors for pressure measurements, the signals collected by the PVDF wire sensor with 2t were compared with those by the hydrophone in both the time and frequency domains, as plotted in Figure 8 and Figure 9, respectively. Figure 8 demonstrates the same oscillatory behavior of the magnitudes of the leak noise signals with a slightly higher level for the wire sensor. Almost the same ASD levels can be found in Figure 9, in particular below 500 Hz. Above 500 Hz, the noise floors for two types of sensors were reached. As can be seen from these figures, the pressure sensitivity of the PVDF wire sensor with 2t was slightly higher than that of the hydrophone. The pressure sensitivity of the hydrophone was 0.12 pC/Pa. Substituting of the pipe and PVDF wire parameters into Equation (11), the sensitivity was calculated to be 0.19 pC/Pa for the wire sensor with 2t. There was a good agreement of pressure sensitivities between the predicted and the measured results. Moreover, the leak signals in the frequency domain demonstrated the low-pass behavior, since the amplitudes of leak signals given by Equation (13) attenuated exponentially with frequency. 

Next, the leak noise signals collected by PVDF wire sensors with 2t, 4t, and 6t were used in the analysis to verify the theory in Section 2. The signals measured by the sensors were compared in both the time and frequency domains. As can be seen from Figure 10, the voltage measured by the wire sensor was roughly proportional to the winding number (with slight derivations of the measured data due to the locations of the wire sensors). Figure 11 plots the corresponding ASDs. The ASD plots exhibited similar trends with increases of roughly 6 dB and 9.5 dB for the sensors with 4t and 6t, compared to 2t in the frequency range of interest. This confirms the theoretical findings and indicates that the pressure sensitivity of the proposed wire sensors can be enhanced by adopting more winding numbers. The theory with experimental validations provides the basis for leak localization using the PVDF wire sensors. However, for practical measurements, potential issues need to be accounted for, and this will be discussed in the next section. 

### 4.2. Leak Localization 

#### 4.2.1. In-Air Case

As mentioned previously, the leak noise signals were measured by the PVDF wire sensors with 6t and hydrophones. Frequency analysis was conducted on the high-pass leak noise signals. Figure 12a plots the coherence functions of the leak noise measured. In general, the coherence was poor in the frequency range up to 1 kHz for both types of sensors. A further check on the phase spectrum revealed that the phase was not in a linear relationship with the frequency. This confirms that the measurements of leak noise were not straightforward. as illustrated in Figure 3, for leakage detection. The reason for this was that the water pipe in the laboratory was rested on three supports, thereby affecting the vibrational behavior of the pipe.

#### 4.2.2. In-Water Case

In this section, the water pipe was submerged and the leak noise signals were subsequently measured by using a pair of PVDF wire sensors with 6t and hydrophone pair at the distances of 0.1 m and 0.9 m. To compare the performance, the time-domain signals were firstly normalized, with respect to the respective maximum magnitudes for each sensor. 

Figure 13 shows the normalized time-domain leak signals at 0.9 m from the leak hole. It is clear that the oscillatory behavior of the signals measured by the PVDF wire sensor and hydrophone was almost identical. The ASDs of the corresponding sensor signals are plotted in Figure 14. Again, similar trends can be found in the leak noise measurements.

Figure 15 shows the coherence function and the phase spectrum of a pair of PVDF wire sensors and a pair of hydrophones at 0.1 m and 0.9 m. It can be seen from the Figure that the coherence of both sensors were improved in comparison with the in-air case, as plotted in Figure 12a, in the entire frequency range up to 1 kHz. Furthermore, as shown in Figure 15b, the phase varied approximately linearly with frequency, as anticipated. Based on the slopes of the unwrapped phase, the time delays were calculated to be 0.0039 s and 0.0042 s for signals measured by hydrophones and PVDF wire sensors, respectively. It was not possible to accurately predict the propagation wave speed due to the uncertainties in the elastic properties of the PE pipe and variations, with respect to temperature. The corresponding wave speed was measured on-site with detailed information given in [36], which was found to be 209 m/s at the time of testing. The calculated leak locations were 0.908 m and 0.940 m relative to the further hydrophone and PVDF wire sensor. Noting that there was a gap of 3 cm between the two sensor types due to installation, accurate leak localization was achieved by both PVDF wire sensors and hydrophones. 

In summary, the PVDF wire sensors proposed here have shown that if good adhesion between the sensor and the pipe wall is achieved (i.e., in the absence of slippage at the pipe wall/wire interface), and no pipe bulging occurs, the pressure sensitivity can be considerable for plastic pipes. This opens up new possibilities for the detection of water leaks in plastic pipes. However, in practical applications, a limitation for such sensors is that both aforementioned scenarios involving slippage and pipe bulging are likely to occur, particularly when the adhesives degrade over time, thus potentially reducing the pressure sensitivity dramatically. To the authors’ knowledge, their effects on the pressure sensitivity, to date, cannot be quantifiable. Further research is required to evaluate the performance for sensing the leak noise signal in real pipeline network. 

## 5. Conclusions

In this paper, the use of the PVDF wire sensors has been investigated for pipeline leakage localization. The performance of the wire sensor has been studied with respect to pressure sensitivity for monitoring the pipe vibration due to leakage. Leak noise measurements have been performed in a PE water pipe in the laboratory under controlled conditions. Theoretical analysis with experimental validations has confirmed the effectiveness of the wire sensors for the detection and location of pipeline leakage by comparing the performance of the PVDF wire sensors and commercial hydrophones. 

Two sets of measurements have been made, associated with pressure sensitivity and leak localization on the in-air and in-water pipe rig. From the measurements of pressure sensitivity on the in-air pipe, it has been demonstrated that good pressure sensitivity can be achieved by the selected PVDF wire sensors with possible improvement over the sensitivity by adopting higher winding numbers. To overcome the undesirable effects of the supports on the wave behavior, leak localization measurements have been done on the in-water pipe. Frequency analysis has clearly shown the effectiveness of the wire sensor for accurate location of the pipe leakage. Construction of a buried PE water pipe is under way, and more measurements in practical situations will be reported at a future date. This work may offer an alternative to leak noise transducers due to it easy employments and low cost. 

## Figures and Tables

**Figure 1 sensors-20-00692-f001:**
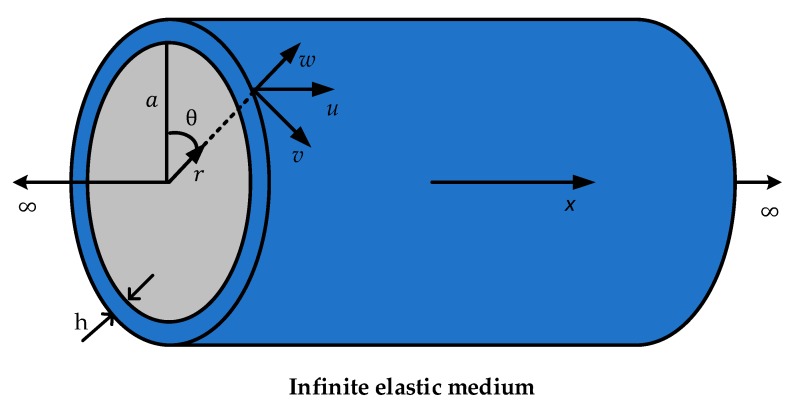
The coordinate system for a fluid-filled pipe in an infinite elastic medium.

**Figure 2 sensors-20-00692-f002:**
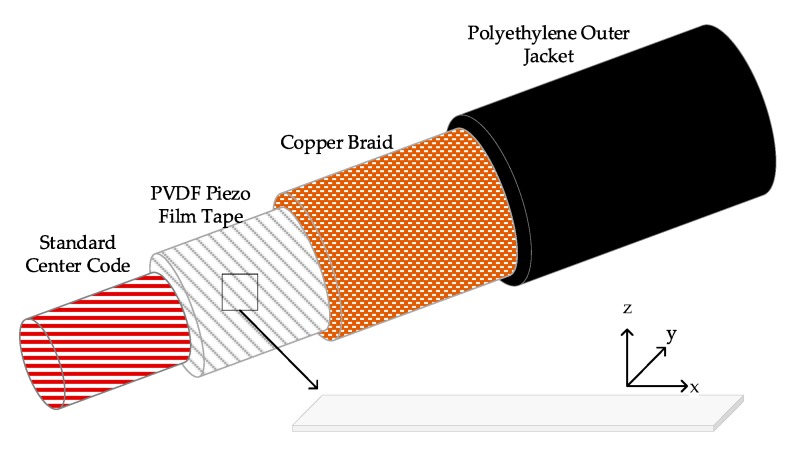
Schematic design of the polyvinylidene fluoride (PVDF) wire sensor.

**Figure 3 sensors-20-00692-f003:**
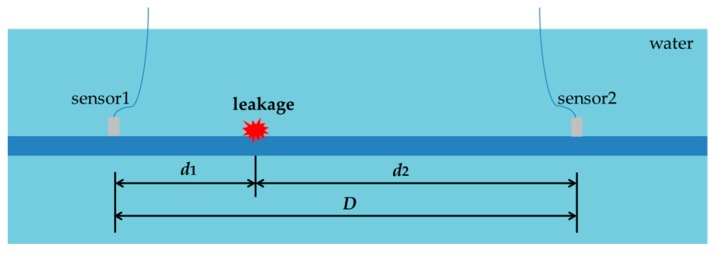
Schematic diagram showing sensor deployment for pipeline leakage detection.

**Figure 4 sensors-20-00692-f004:**
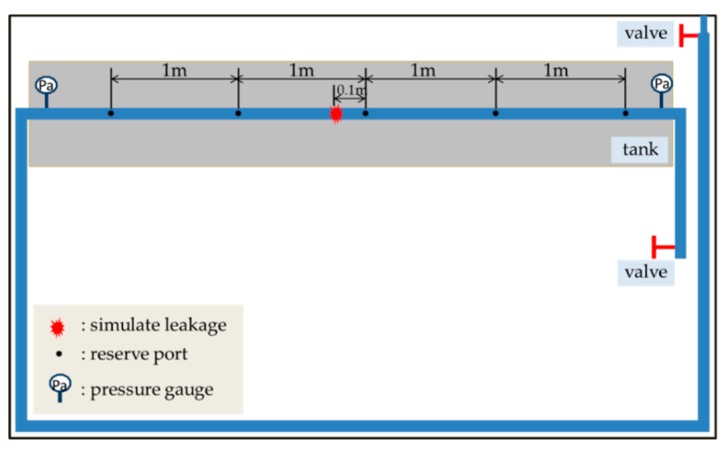
Schematic diagram of the test pipe system.

**Figure 5 sensors-20-00692-f005:**
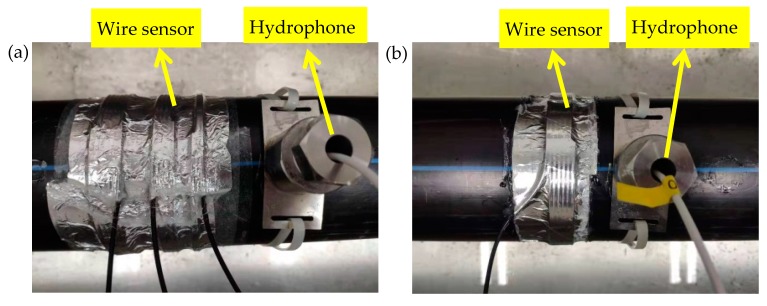
Sensor installation: (**a**) for sensitivity measurements; (**b**) for leak localization measurements.

**Figure 6 sensors-20-00692-f006:**
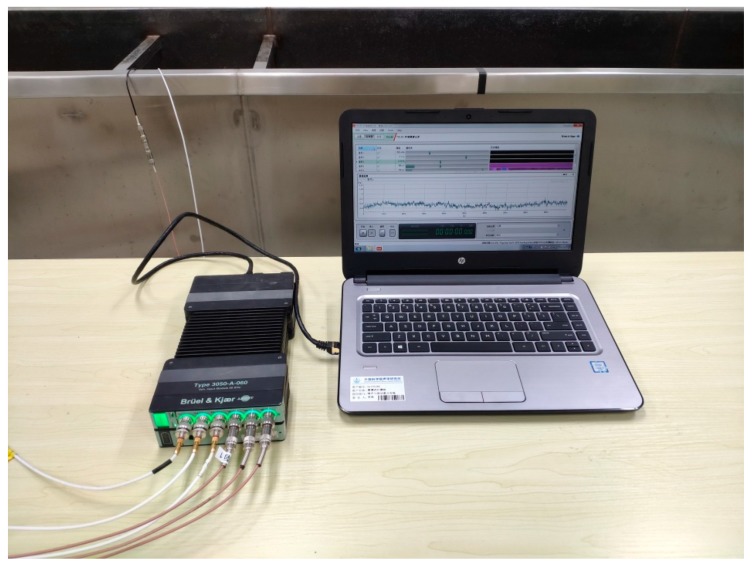
Data acquisition system.

**Figure 7 sensors-20-00692-f007:**
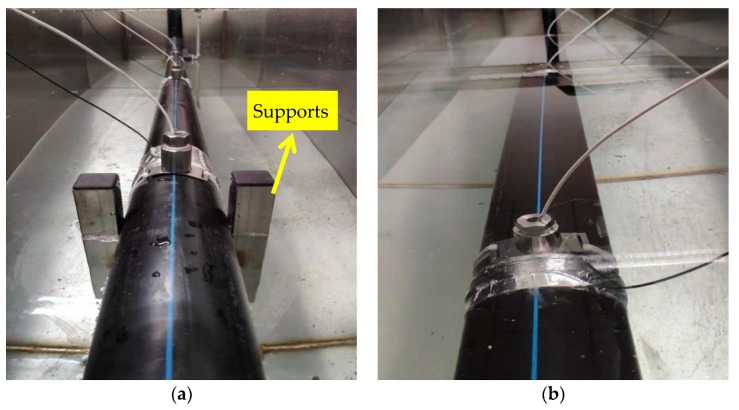
Test rig: (**a**) on supports; (**b**) submerged.

**Figure 8 sensors-20-00692-f008:**
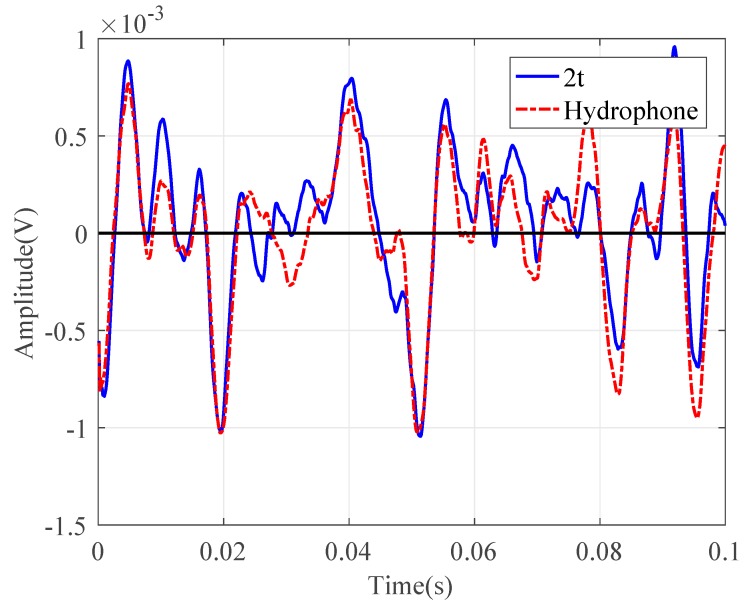
Time-domain leak noise signals collected by the PVDF wire sensor with two-turn (2t) and hydrophone.

**Figure 9 sensors-20-00692-f009:**
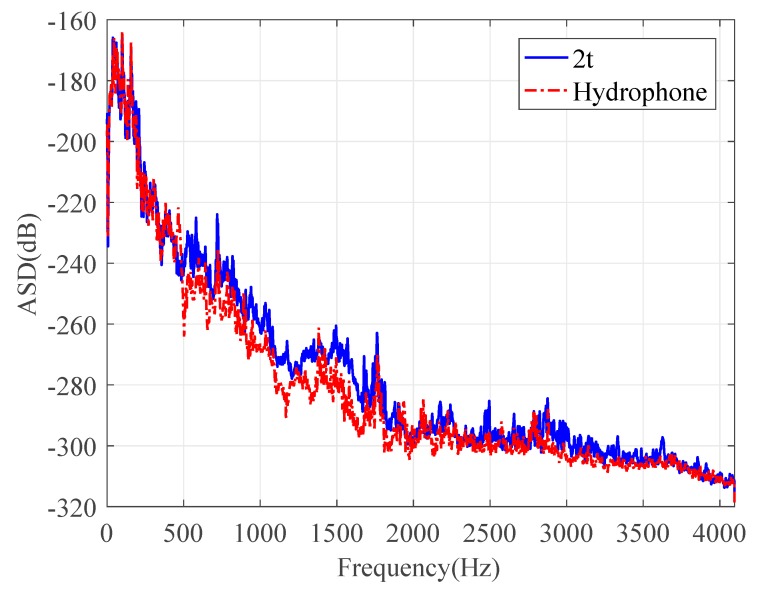
Auto-spectral densities (ASDs) of the leak noise signals collected by the PVDF wire sensor with 2t and hydrophone.

**Figure 10 sensors-20-00692-f010:**
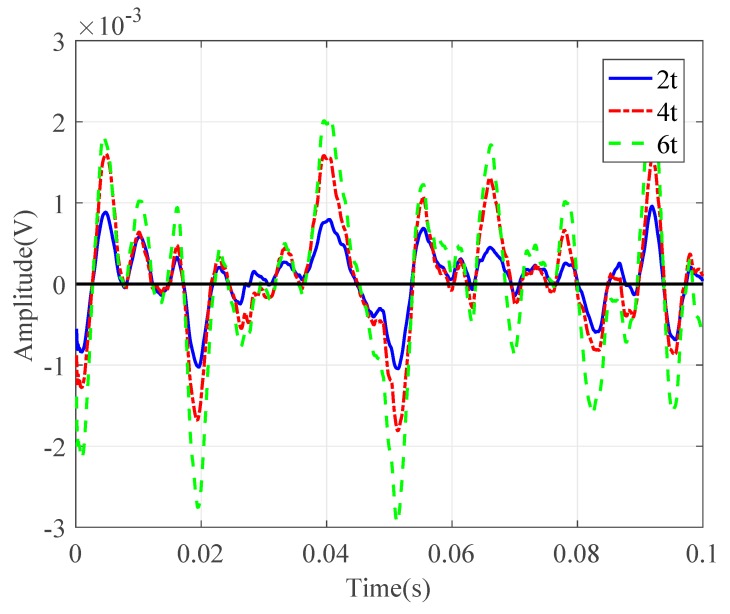
Time-domain leak noise signals collected by PVDF wire sensors with different winding numbers.

**Figure 11 sensors-20-00692-f011:**
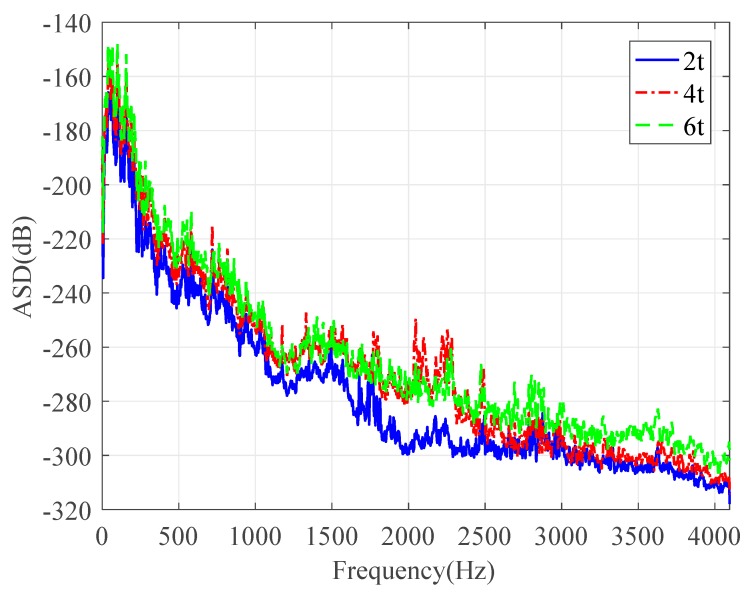
ASDs of leak noise signals collected by PVDF wire sensors with different winding numbers.

**Figure 12 sensors-20-00692-f012:**
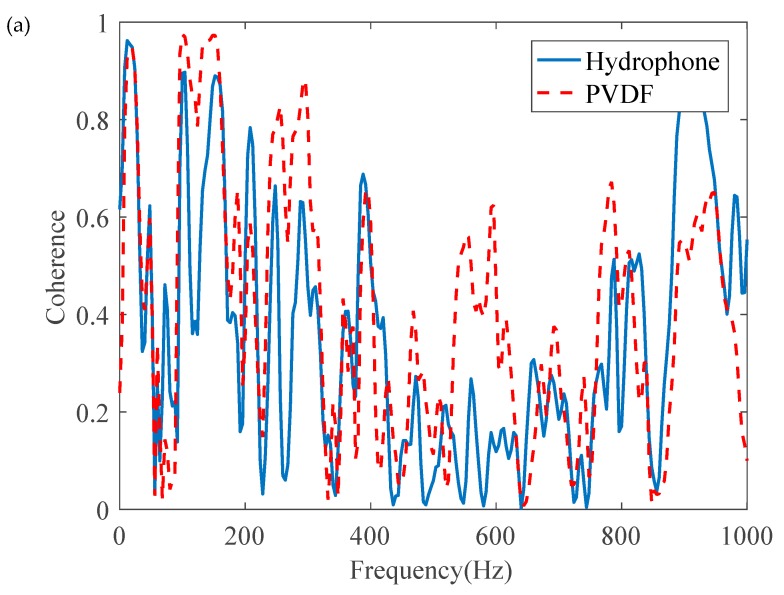
Frequency results of a pair of PVDF wire sensors and a hydrophone pair at the distances of 0.1 m and 0.9 m from the leak hole (in-air case): (**a**) coherence; (**b**) phase spectrum.

**Figure 13 sensors-20-00692-f013:**
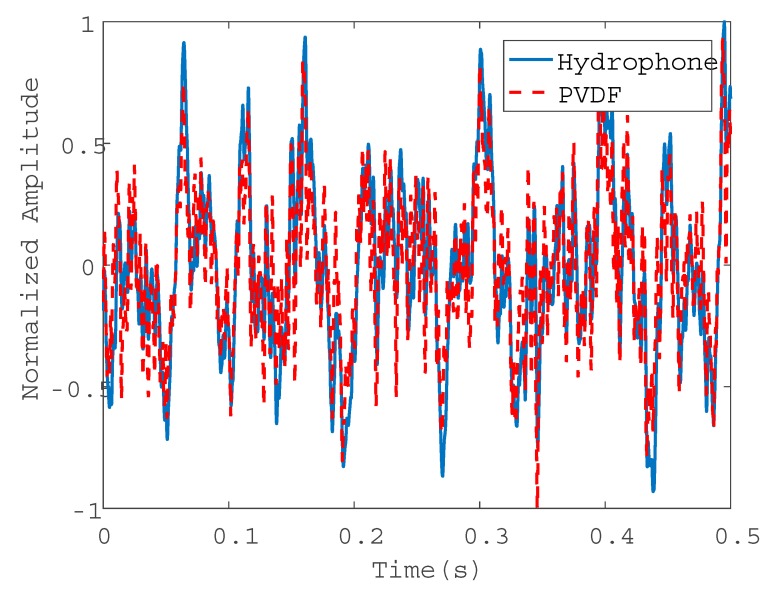
Time-domain leak signals collected by the PVDF wire sensor of 6t and hydrophone at 0.9 m (in-water case).

**Figure 14 sensors-20-00692-f014:**
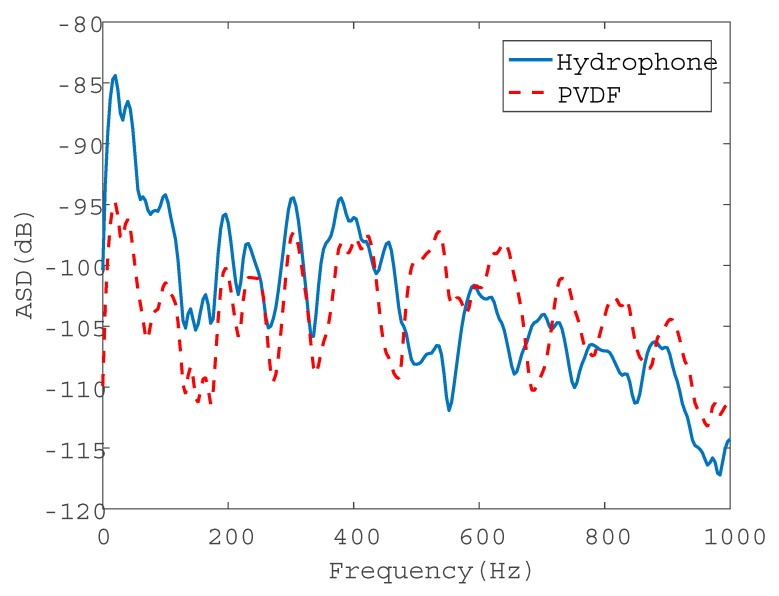
ASDs of the leak noise signals collected by the PVDF wire sensors of 6t and hydrophone at 0.9 m (in-water case).

**Figure 15 sensors-20-00692-f015:**
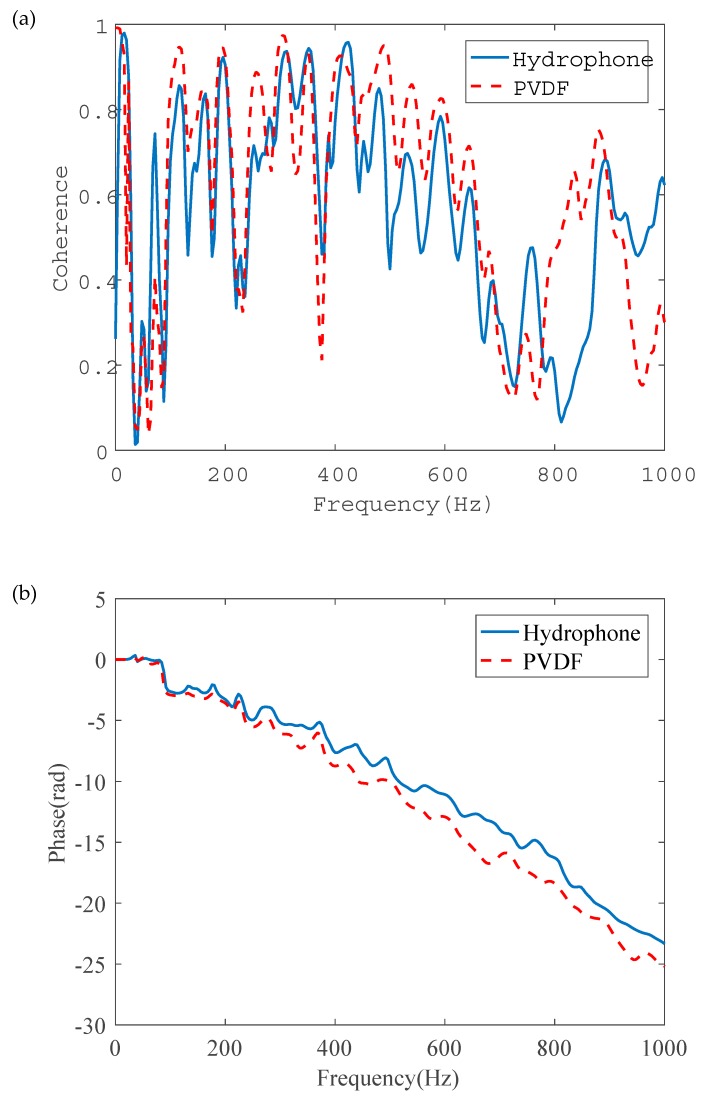
Frequency results of a pair of PVDF wire sensors and hydrophone pair at the distances of 0.1 m and 0.9 m (in-water case): (**a**) coherence; (**b**) phase spectrum.

**Table 1 sensors-20-00692-t001:** Properties of the cast iron and polyethylene (PE) pipes.

Material	Cast Iron	PVC
Density ($\mathrm{kg}/m^{3}$)	7100	2000
Young’s Modulus ($\mathrm{GN}/m^{2}$)	100	5
Poisson’s Ratio	0.29	0.4
Radius (mm)	50	50
Thickness (mm)	5	5

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
