# Peer review of "Use of PVDF Wire Sensors for Leakage Localization in a Fluid-Filled Pipe"

_sensors, 2020, doi:10.3390/s20030692_

Round 1

Reviewer 1 Report

This paper was well written and organized in its content and structure. The method development and application were also well performed in this study. Overall this paper is in very good quality in terms of technical method development and practical applications. 

In this study, the PVDF wire sensor was developed and applied for leak detection, which is significant for the community of urban water distribution analysis and management. Actually, as mentioned by the authors in the paper, leak detection (or water loss control) in urban water distribution system is a urgent need in current practice, for its economic and societal impacts. In this regard, this paper presents a timely study for improving/enhancing/supplementing the existing method to solving this issue. 

As a researcher in this similar topic/field (i.e., leak detection and water resources management), I personally would like to see this paper published. But as a reviewer, I still want to raise some concerns so as to improve this paper quality (or to provide some optional suggestions to the authors for their excellent work):

(1) I agree with the authors that many of current methods are mainly developed for metallic pipe materials, while very few for the plastic pipes such as PE or PVC pipes that are commonly used in urban water pipe systems (supply or drainage). So, from this perspective, the authors are suggested to review more thoroughly the studies in the literature about the leak detection or diagnosis in plastic pipes, which I think will be more helpful to highlight the contribution of current work. For instance, but not limited to: 

     (a) Duan, H.F., et al. "System response function based leak detection in viscoelastic pipeline," Journal of Hydraulic Engineering – ASCE, 138(2), 143-153, 2012;

     (b) Martini A., et al. "Leak Detection in Water-Filled Small-Diameter Polyethylene Pipes by Means of Acoustic Emission Measurements", Applied Sciences, 7(2), 2017.

     (c) Bracken M. et al. "Transmission Main and Plastic Pipe Leak Detection Using Advanced Correlation Technology: Case Studies", Pipelines Conference 2011 July 23-27, 2011 | Seattle, Washington, United States.

     (d) ...

(2) According to the various experience of my team work, the leak detection problem biomes more difficult and critical in plastic pipes than in metallic pipes. One of main reasons may be attributed to the uncertainty and unknown of the parameters and coefficients of such plastic pipes. Specifically, it becomes more difficult to obtain an accurate parameters of plastic pipes such as wave speed and pipe-wall material properties (E, e, rho, ... in Eqs. 6 and 7, etc.), especially for those pipes being used for many years. To this end, it is critical to identify such parameters and properties of plastic pipes prior to the use and application of the leak detection method and apparatus in practice. However, I also understand that how to identify and evaluate the plastic pipe properties is not the exact scope of current paper. In this connect, I would like to suggest the authors could conduct a brief discussion of this issue for their method application and results analysis (for example, some recent publications about the analysis of viscoelastic parameters of plastic pipes can be found in several professional journals: e.g., ASCE Journal of Hydraulic Engineering): 

(3) The advantages of the developed sensors and method for leak detection have been well illustrated in the paper, through the comparison with the traditional Hydrophone-based method. I would like to suggest the authors would discuss the possible/potential limitations of their developed device and method for practical applications, which will be beneficial to others in this community. Again, I think this will be optional to the authors.

In summary, this paper is very good and ready for publishing on this journal, but I strongly suggest the authors could address such minor issues in their final version for publication. Congratulations to the authors for this good work.

Reviewer 2 Report

The paper deals with a novel sensor for detecting water leaks in pipeline networks. Its effectiveness and reliability are assessed through experimental comparison with hydrophones. The following comments should be addressed.

MAIN REMARKS

1) Information on the connection between the PVDF wire sensors and the frontend does not appear exhaustive. Do the sensors require a charge amplifier? Please provide further details on the sensor setup.

2) Section 3.2. It appears that ensuring a very good adhesion between the PVDF wire and the pipe is essential for achieving reliable measurements. However, in practical applications, this may represent a limitation for such sensors, particularly when degradation over time of the adhesives occurs. The Authors should include in the paper some considerations on this potential issue.

3) Section 4.1. According to the Authors’ statement, the PVDF wire sensor are slightly more sensitive than hydrophones. Actually, the sensitivity of the adopted 2t sensor appears SIGNIFICANTLY higher (about +58%). Nonetheless, the amplitudes of the measured signals appear comparable (Fig. 9), hence implying that the actual sensitivity of the PVDF wire sensor is lower than expected. The statement should be reformulated and further considerations and clarifications on these aspects should be added to Section 4.1.

4) Figure 8. For the sake of completeness, it would be advisable extending the x-axis of the ASDs over the available frequency range (i.e. up to 4096 Hz), at least in this figure, in order to appreciate that no significant frequency content is present above 1 kHz.

5) The motivation for using time-frequency analyses on steady-state signals is unclear. Did the authors expected any transient phenomena? Apparently, Fig. 14 and the corresponding description (lines 264-267) do not add any useful information and may be omitted.

MINOR COMMENTS

6) Eq. (1). The quantity J0' is not mentioned in the description of the equation terms.

7) Section 2.2, lines 126-127, “[..] its piezoelectric coefficients are much higher than other piezoelectric materials.” For the sake of completeness, it would be advisable quantifying such coefficients.

8) Section 4.1, lines 220-221, “Moreover, the leak signals in the frequency domain demonstrate the low-pass behavior.” Actually, in Section 3.1 it is declared that the signals are pre-processed with a high-pass filter. Please, clarify.

9) For a better overview on the latest advances on the topic of leak detection in water distribution networks, additional recent works different from the ones of the Authors should be mentioned. To this purpose, the following recent papers may be included in the References as examples.

- Martini, A.; Rivola, A.; Troncossi, M. Autocorrelation Analysis of Vibro-Acoustic Signals Measured in a Test Field for Water Leak Detection. Applied Sciences 2018, 8(12):2450, 1-15. doi:10.3390/app8122450

- Liu, Y.; Ma, X.; Li, Y.; Tie, Y.; Zhang, Y.; Gao, J. Water pipeline leakage detection based on machine learning and wireless sensor networks. Sensors 2019, 19(23):5086, 1-21. doi: 3390/s19235086

- Yazdekhasti, S; Piratla, K.R.; Sorber, J.; Atamturktur, S; Khan, A.; Shukla, H. Sustainability Analysis of a Leakage-Monitoring Technique for Water Pipeline Networks. Journal of Pipeline Systems Engineering and Practice 2020, 11(1):04019052, 1-16. doi:1061/(ASCE)PS.1949-1204.0000425

Round 2

Reviewer 1 Report

The paper has been improved by addressing the comments. Now I recommend it is acceptable for this journal.

Reviewer 2 Report

The review comments have been addressed quite satisfactorily.